# Local resource availability drives habitat use by a threatened avian granivore in savanna woodlands

**John M. van Osta**[1,2]*, **Brad Dreis**[1,2], **Laura F. Grogan**[1], **J. Guy Castley**[1]

**1** School of Environment and Science, Griffith University, Queensland, Australia, **2** E2M Pty Ltd, Milton, Queensland, Australia

* john.vanosta@griffithuni.edu.au

**Data Availability Statement:** Data and code supporting this paper are shared through an open access repository: https://doi.org/10.6084/m9.figshare.26043391. Sensitive locational information for the SBTF has been excluded as the

## Abstract

Conserving threatened species relies on an understanding of their habitat requirements. This is especially relevant for granivorous birds, whose habitat use and movement patterns are intricately linked to the spatial and temporal availability of resources such as food and water. In this study, we investigated the habitat use, home range and daily activity patterns of the Endangered Southern Black-throated Finch (SBTF; *Poephila cincta cincta*) within a 75,000 ha savanna woodland study area in northeastern Australia. This semi-arid region is one of the key remaining strongholds for the species and is characterised by substantially different habitat and climatic conditions than areas where previous research on this species has been undertaken. We radio tracked 142 SBTF using both manual tracking and an array of 27 automated radio towers, which revealed a strong preference for eucalypt-dominated grassy woodland communities. The preference for these habitats also increased with decreasing distance to permanent water. While SBTF occupied large home ranges, individual SBTF were largely sedentary during the radio tracking period (21.8, range = 0.83–120 days), with few landscape-scale movements of more than 4.5 km. Daily foraging activity of SBTF exhibited bimodal peaks in the early morning and late afternoon, while other activities were greatest from the late morning to the early afternoon. Compared to other estrildid finches, our research suggests that SBTF track resources at a local scale across a large home range. We postulate that in times of resource scarcity SBTF may use dietary diversification, instead of landscape or regional-scale nomadic movements, to meet their resource needs. The species' movement patterns underscore the importance of local scale habitat management to facilitate resource availability throughout the year. Furthermore, our research helps target monitoring designs for granivorous birds that focus on the species' diurnal activity patterns.

## Introduction

Habitat use is a key focus of research effort as a species' habitat preferences can ultimately influence survival and population persistence [1]. Understanding an animal's habitat use, home range and movements helps to identify how animals may be influenced by the

species is endangered and listed as a confidential species by the State Government where this research was undertaken (Queensland, Australia). Further access may be requested to the research permit holder at permits@e2m.com.au.

**Funding:** Bravus Mining and Resources provided funding, contributed to study design and supported data collection. The funder had no role in data analysis, decision to publish or preparation of the manuscript.

**Competing interests:** The authors have declared that no competing interests exist.

availability and distribution of resources such as food, water, and breeding habitats [2, 3]. Central to habitat use studies is the analysis of habitat preferences, home range and movement patterns [4]. This knowledge becomes even more critical for species that are threatened and reliant on targeted conservation efforts [5, 6]. In Australia, granivorous birds, especially ground feeders, are disproportionately represented in threatened species lists and have experienced significant range declines due to changes in resource availability [7, 8]. These declines are primarily attributed to the spread of pastoralism, which leads to land clearing, altered vegetation composition, changed fire regimes, increased grazing pressure, as well as predation by introduced species [9, 10].

The granivorous bird assemblage in the tropical savanna biome of northern Australia is rich in estrildid finch species (n = 18), many of which have overlapping distributions [11]. These sympatric species exhibit a wide diversity of mobility and dietary behaviours that reduce their interspecific competition [12, 13]. Species range from Double-barred and Crimson Finches (*Stizoptera bichenovii* and *Neochmia phaeton*) that occupy small home ranges and are dietary generalists through to Gouldian Finch (*Erythrura gouldiae*) and Pictorella Mannikin (*Heteromunia pectoralis*) that are highly nomadic dietary specialists [12, 13]. Understanding the movement, habitat use and activity patterns of these species is important for identifying the resource niches that are required by species in order to appropriately target management actions. This is particularly important for three of the savanna-inhabiting estrildid finches that are listed as threatened at a national level, namely Gouldian Finch, Eastern Star Finch (*Neochmia ruficauda ruficauda*) and Southern Black-throated Finch (*Poephila cincta cincta*) [14]. Understanding the movement and habitat use patterns of these species helps to inform conservation actions, such as the management of fire and grazing that are key habitat modifiers in the savanna woodlands [15–17].

The Southern Black-throated Finch (SBTF) is one of the nationally threatened estrildid finches that has suffered extensive range decline since European settlement and is listed as Endangered under Federal and State legislation [18, 19]. The species inhabits open grassy woodlands [20, 21], and is primarily granivorous [18, 20]. Its habitat has also been subjected to extensive degradation, primarily due to the spread of pastoralism and associated changes in land management practices, including habitat clearing, altered fire regimes, changes in grazing pressure and the spread of introduced grasses [9, 18].

The contemporary SBTF population is concentrated in two remaining strongholds, namely, the Townsville Coastal Plains biogeographic subregion and the Desert Uplands Bioregion [18, 19]. Published movement and habitat use research on SBTF has focused primarily on the population located on the Townsville Coastal Plains [6, 18], while similar research from the Desert Uplands Bioregion is sparse. Importantly, the two areas exhibit substantially different climate and habitat characteristics and could lead to different patterns of habitat use by SBTF in the Desert Uplands. The Townsville Coastal Plains is within the tropical savanna with a dry winter Köppen-Geiger climate zone, while the Desert Uplands Bioregion is within the hot semi-arid climate zone [22]. In addition, the two regions share little overlap in vegetation communities [23], and the Desert Uplands has less than half the area of freshwater wetlands and artificial dams [24, 25].

The aim of this study was to identify the habitat use, movement and diel activity patterns of SBTF within the Desert Uplands Bioregion and to test for seasonal variation in these factors. This study builds on previous research from the Townsville Coastal Plains by Rechetelo et al. [6], and increases the sample size of tracked SBTF and the period over which tracking is undertaken. More broadly, our research aims to improve our understanding of the ecology of the SBTF and aims to contribute to conservation planning and monitoring for the Endangered SBTF and other granivorous birds with similar threatening processes.

## Materials and methods

### Study area

We conducted this study in a 75,000 ha section of the Moray Downs property within the Desert-Uplands Bioregion of Queensland, Australia (Fig 1). The study area receives an average annual rainfall of 541 mm, 80% of which occurs between November and April, marking a characteristic wet season [26]. Land use within the study area includes a combination of native vegetation managed for conservation as an environmental offset, low-intensity cattle grazing and an active coal mine. The majority of permanent water sources are artificial bores and dams that were established for cattle grazing as well as permanent water associated with a major watercourse, the Carmichael River (Fig 1).

Remnant vegetation covers approximately 79% of the study area and includes 10 vegetation communities that we assigned based on similarities in dominant canopy species and soil attributes (Fig 1 and S2 Table). These vegetation communities are based on the Queensland Government's regional ecosystem framework [27], which was extensively ground-truthed

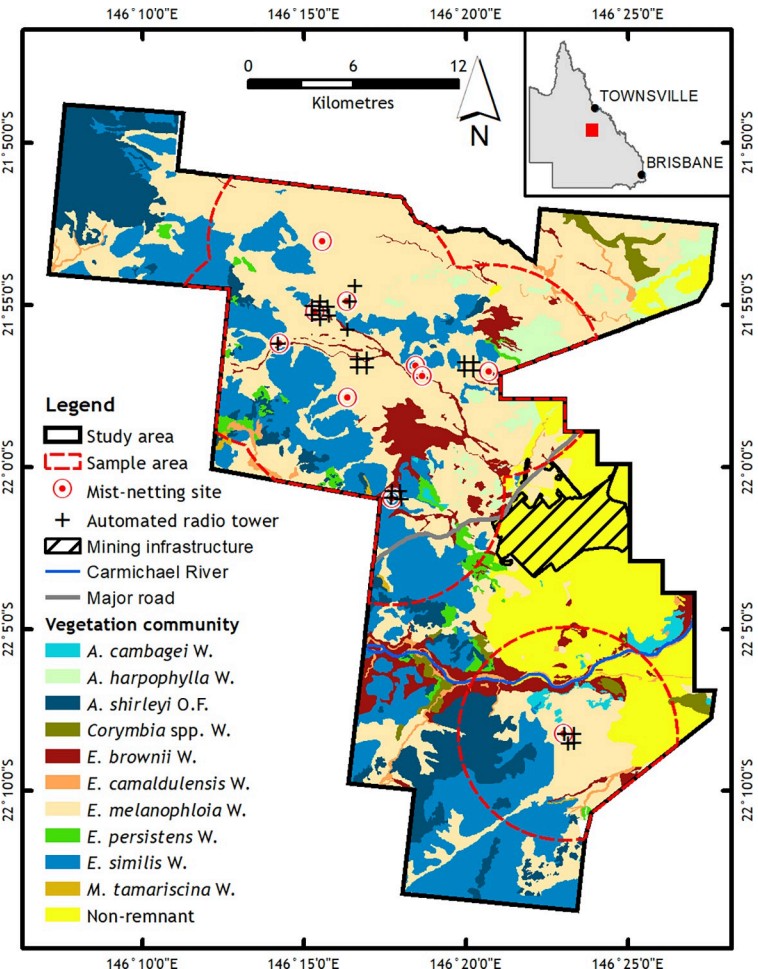

**Fig 1. Study area for Southern Black-throated Finch research in savanna vegetation within the Desert Uplands region of Queensland, Australia.** The map depicts the locations of automated radio towers, mist-netting sites, vegetation communities, major roads, major watercourses and mining infrastructure. Abbreviations include A = *Acacia*, E = *Eucalyptus*, M = *Melaleuca*, O.F. = open forest and W = woodland.

throughout the study area. Broadly, the study area's vegetation comprises semi-arid savanna woodlands that are dominated by eucalypt and Acacia species [17]. Eucalypt woodlands account for 71% of the remnant vegetation, with the most common communities being those dominated by silver-leaved ironbark (*Eucalyptus melanophloia*) (39%) and yellow jacket (*E. similis*) (22%) on deep red earths (Fig 1 and S2 Table). Other remnant habitats include lance-wood (*Acacia shirleyi*) on skeletal Tertiary plateaus, *E. camaldulensis* and *E. brownii* woodlands on alluvial plains associated with watercourses and drainage features, brigalow (*A. harpophylla*) and gidgee (*A. cambagei*) open forests on clay plains, and small areas of *Melaleuca* dominated shrublands (Fig 1 and S2 Table). In addition, non-remnant areas, which have been cleared of native vegetation [27], cover 21% of the study area. Of the non-remnant areas, 16% are used for cattle grazing pastures and 5% are occupied by mining infrastructure and roads (Fig 1).

## Ethics statement

Mist-netting, banding and radio tracking were conducted under the animal ethics permit CA 2020/07/1392, issued by the Queensland Department of Agriculture and Fisheries; research permit WA0025814, issued by the Queensland Department of Environment, Science and Innovation; and Australian Bird and Bat Banding Scheme authority 2832–01, issued by the Australian Government Department of Climate Change, Energy, the Environment and Water.

## Mist-netting and transmitter attachment

Nine mist-netting and radio tracking surveys were conducted between August 2020 and May 2023, with approximately three per year to align with the dry season (May to November), early wet season (December to February) and wet season (February to April). Mist-netting was conducted at 10 locations where SBTF were observed, which included permanent and ephemeral water sources and a single foraging location remote from water. A total of 142 SBTF were fitted with Lotek NTQB2–2 Nanotag radio transmitters, which weighed 0.32 g, approximately 2.1% of the average SBTF weight. An average of 15.8 ± 5.59 transmitters were applied per survey period ($\bar{x}$ ± 1 SD, range = 7–23) from 4.89 ± 1.27 independent locations ($\bar{x}$ ± 1 SD, range = 3–7) per survey. Across the entire study period 14.2 ± 11.0 transmitters were applied per survey location ($\bar{x}$ ± 1 SD, range = 1–38) (S1 Table). Transmitters were attached to the scapular region of the birds by trimming a small patch of feathers level with the skin and using cyanoacrylate glue [6, 28, 29]. Prior to release, transmitter function and signal detection were tested using a handheld receiver (Lotek SRX1200 M2). All of the radio transmitters were set to 150.6 MHz and emitted uniquely coded radio pulses, allowing multiple transmitters to be recorded simultaneously using both manual tracking as well as the automated radio telemetry arrays. Transmitters with a 13 second pulse interval (n = 73) and 97 day battery life were initially used based off an estimated transmitter retention time [29]. Following initial field surveys and evaluation of tag retention times, we changed to a 3 second pulse interval and 29 day battery life (n = 69) to better reflect the tag retention that was being achieved, which was 21.8 ± 22.5 days ($\bar{x}$ ± 1 SD, range = 0.83–120 days) and improve manual radio tracking efficiency. All SBTF captured were also fitted with a uniquely coded Australian Bird and Bat Banding Scheme supplied alloy band.

## Manual radio tracking

We used a combination of tracking approaches by combining manual tracking with an automated radio telemetry system to assess the movement and habitat use of 142 SBTF over a

three-year period. We excluded tracking data obtained within 24 hours of transmitter attachment from all statistical analyses to allow a post-release acclimation period.

We manually radio-tracked SBTF using a handheld 3-element Yagi-Uda antenna connected to a Lotek SRX1200-M2 receiver. We aimed to capture the location of each tracked SBTF approximately twice per day. Where field time allowed, SBTF were also tracked at a higher temporal resolution, of up to one location per 15 minutes until the signal was lost, which provided additional information on diel activity patterns. Tracking bouts ranged from single sightings to approximately 8 hours of tracking per bird. We collected the location of tracked SBTF after visually resighting the individuals, where possible. Care was taken to avoid disturbing the SBTF while radio-tracking to limit behavioural changes associated with tracking. For higher temporal resolution tracking, SBTF positions were occasionally estimated from the maximum radio signal strength and direction when achieving a visual resight was likely to flush foraging birds. We recorded the GPS locations of tracked SBTF using a Trimble TDC600 from the position where the radio-tracked bird was prior to moving off. A positional error of 20 m was assigned to the manual radio tracking locations, approximating the uncertainty of locating the exact position that the radio-tracked SBTF was observed, as well as additional GPS error.

## Automated radio telemetry system

We used an array of 27 automated radio telemetry (ART) towers to remotely and continuously estimate the location of radio tracked SBTF when they were within the range of one or more towers. Site-based testing of the towers found a typical detection distance of approximately 300 m when SBTF were foraging on the ground and 800 m when SBTF were perching, which they typically do 2–10 m above ground height. We installed the ART towers within remnant vegetation in areas of known foraging, drinking and nesting habitats for SBTF. The setup included four arrays of four towers each, and an additional array of 11 towers that comprised seven units the centre surrounded by four units in peripheral areas (Fig 1). This design allowed simultaneous detections of SBTF across multiple towers, particularly in the centre of an array area. The ART towers were on average 10.3 km (range: 5.03–14.9 km) from the nearest mining infrastructure.

Automated radio telemetry data require extensive quality assurance steps to ensure reliable location estimates [30, 31]. Our quality assurance steps included an initial filter of the ART data to include only those tag detections that were recorded after the time of release of the bird and within the maximum possible transmitter battery life of approximately 29 days or 97 days, depending on tag pulse rate. Field observations found that transmitters typically exceeded the manufacturers specification by approximately 10–40% and as such we multiplied the life of the transmitter published by the manufacturer (Lotek) by 1.5, to ensure that all true tag detections were captured. After filtering, we arranged the detections by tower ID and time and manually screened for outliers. The minimum convex polygons (MCP) for each tag ID were calculated using ArcGIS Pro (ESRI), and the outliers were manually verified. We then checked for dropped transmitters by plotting the received signal strength of each transmitter on each antenna and manually screening for a time after which the signal strength did not change, suggesting that the transmitter was stationary for an extended period.

Transmitter positions were estimated from the ART data using a data pipeline that is described in detail by van Osta et al. [32]. The data pipeline used site-specific training data collected manually by radio-tracking the SBTF to develop a model to estimate transmitter positions [32]. The median error of the ART position data was estimated at 235 m using a test dataset, described by van Osta et al. [32], which was factored into the home range estimates discussed below.

## Landscape scale habitat use

Animals select habitat at multiple spatiotemporal scales [33]. We chose to assess SBTF habitat use at the 'landscape scale', which is the area that an individual chooses to inhabit on a time-scale of days to months [33]. This scale is the most suitable for the intensity and sampling duration at which we obtained radio tracking data. The landscape scale is also suitable for the 75,000 ha study area, which represents an area that is much larger than the individual home ranges of SBTF, while being small enough that a SBTF could, if it chose to, access any part of the study area over a timescale of days [33, 34].

To assess landscape scale habitat use, we developed a resource selection function that used presence sites from manual radio tracking and ART locations and generated pseudoabsence sites [35, 36]. Presence sites were randomly subsampled to a maximum of two sites per day from each of the manual radio tracking and ART locations to minimise bias from auto-correlation of radio tracking data [6, 37]. The minimum time allowed between ART subsamples was 15 minutes, which matched the minimum sampling frequency of manual radio tracking. We generated pseudoabsence sites within a subset of the study area that we defined as the 'sample area' (Fig 1) at a 10x multiple of presence sites using ArcGIS Pro (ESRI), which followed recommendations by Barbet-Massin et al. [39]. The sample area was a circle centred on each SBTF trapping site that had a radius equal to the maximum distance that these birds were observed to travel from the trapping site within our study (6.07 km). Areas on adjacent properties within these buffers were excluded due to access restrictions. Since detection of radio-tracked SBTF was likely to be highest near vehicle tracks, which is a common constraint of land-based radio-telemetry studies [38], we randomly generated pseudoabsence sites at the same proportional distribution with distance from vehicle tracks as our presence sites (S1 and S2 Figs) [39]. The coverage of pseudoabsence sites therefore closely approximated the effective radio-tracking area coverage of our study.

The independent variables selected for the model included vegetation community (S2 Table), distance to permanent water and season. We mapped the distributions of permanent water sources with a high level of confidence using a combination of Queensland Government mapping, information supplied by landholders, interpretation of high-resolution aerial imagery and manual verification during field surveys. Additionally, we tested the effect of season on habitat use by adding first order interaction effects between season and vegetation community and season and distance to permanent water. The study area's 'wet season' typically occurs between November and April; however, there is year to year variability in the onset and cessation of rainfall that defines the wet season. We therefore categorised records as wet or dry season based on the yearly timing of rainfall, availability of ephemeral water and growth of annual grasses.

We used Firth's logistic regression as the resource selection model, which was implemented in the logistf R package [40]. Firth's logistic regression was selected due to complete separation of the response variable (SBTF presence/pseudoabsence) among some vegetation communities. Firth's method is robust to convergence issues in the presence of separation while still producing unbiased model estimates [41, 42]. We converted model estimates to odds ratios using the sjPlot package [43], which is a standard approach to facilitate interpretation of logistic regression coefficients [44].

## Home range

We calculated autocorrelated kernel density estimate (AKDE) home ranges using the ctmm package version 1.2.0 [45]. We used the pHREML wAKDE$_c$ method, following recommendations of Silva et al. [46], which applied bias correction methods for inconsistent sampling

intervals, autocorrelation of locations and small sample sizes. For each transmitter, we initially trialled the inclusion of all manual radio-tracking and ART location estimates within the model. However, despite the autocorrelation corrections applied by the model, we found that the home ranges were heavily biased to the ART data. To account for this, we subsampled the ART location estimates to two random locations per day, with a minimum of 15 minutes between subsamples, which approximated the sampling frequency of the manual radio tracking dataset.

We included estimated positional errors for each location in the AKDE home range calculations. These positional errors were 20 m for manual radio tracking locations and 235 m for ART locations, which are described in more detail in the respective methods sections above. Home ranges and core ranges were estimated using the 95% and 50% AKDE contours, respectively [47]. Home range estimates were plotted against their effective sample sizes, which are the number of statistically independent samples within the data [48]. We used these plots to verify that the autocorrelation and sample size correction methods were sufficient to avoid small effective sample sizes underestimating the SBTF home ranges [46].

We tested the impact of season on the 95% AKDE home range of SBTF using a generalised linear mixed effects model with a log link gamma distribution using the glmmTMB package version 1.1.7 [49]. Wet and dry seasons were categorised following the same method as the landscape scale habitat use method. The mist-netting site was added as a random effect to this model.

We also calculated minimum convex polygons for each transmitter using ArcGIS Pro (ESRI), which is a coarse home range tool that we selected to provide a comparative method to SBTF home range estimates for the Townsville Coastal Plains population produced by Rechetelo et al. [6].

## Diel activity patterns

Where possible during manual radio tracking, observers categorised SBTF activity as foraging, drinking, within a nest, flying, perching or preening. The flying, perching and preening categories were combined for statistical analysis as these activities were often undertaken in association with each other.

We used the activity package version 1.3.4 [50] to standardise the time to sunrise and sunset using a double anchoring method [51]. This method anchors the time to equinox sunrise and sunset times for the study area [51]. We transformed the clock times so that equinox sunrise occurred at 6 am and sunset occurred at 6 pm. Anchoring time to both sunrise and sunset is important for SBTF, which arrive and depart their nest based on sunrise and sunset times [52].

There was unequal survey effort across seasons and times of day, making a direct analysis of counts inappropriate. We instead created an unbiased metric for activity by calculating the proportion of each activity relative to other activities for categorical combinations of season and time of day. The time of day was assigned to five categorical bins, which included early morning (6:30–8:30 am), late morning (8:30–10:30 am), midday (10:30 am–1:30 pm), early afternoon (1:30–3:30 pm) and late afternoon (3:30–5:30 pm). Night and twilight hours (within 30 minutes of dawn and dusk) were excluded due to a limited sample size of manual tracking records. Wet and dry seasons were categorised following the same method as the landscape scale habitat use method. We then used a linear model to test the impact of time of day and season on the proportion of observations made for each activity. The model included interaction effects between activity and time of day, as well as between activity and season. All data analyses were undertaken using R statistical language version 4.3.0 [53]. Unless otherwise specified, data in the results section below are reported as mean ± SD (range).

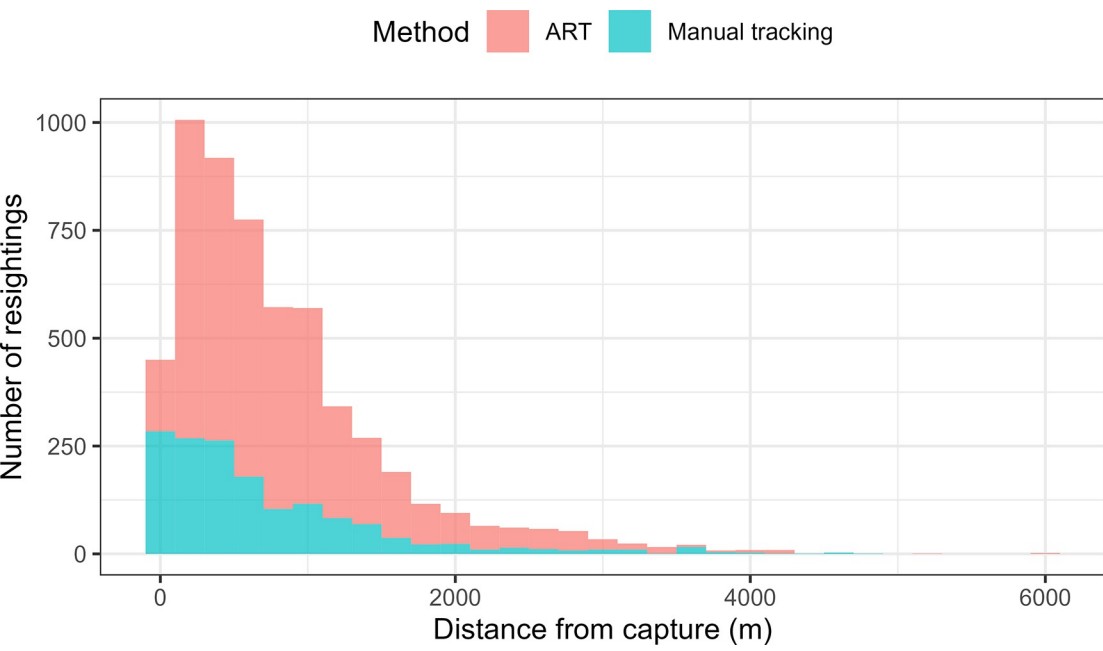

**Fig 2. Histogram showing the distribution of radio tracked Southern Black-throated Finch in the Desert Uplands region of Queensland, Australia.** Resightings are relative to the distance from their point of capture and include resightings from automated radio telemetry (ART, red) and manual radio tracking (blue).

## Results

### Radio tracking

For each of the 142 radio tracked SBTF, we recorded an average of 1890 ± 4180 ART locations (range = 0–45600) and 11.9 ± 11.9 manual radio tracking locations (range = 1–57). All SBTF were detected through manual tracking at least once, while eight SBTF were not detected on the ART receivers post-release. The average number of manual radio tracking detections per day that the individual was detected was 3.3 (range = 1–30).

Following the data subsampling steps described in the methods, which aimed to ensure independence of the data, we used an average of 39.9 ± 32.4 locations per SBTF (range = 5–199) to estimate home ranges and 33.1 ± 29.5 locations per SBTF (range = 2–198) to test landscape scale habitat use.

Radio tracked SBTF were resighted at an average of 810 ± 730 m from their initial capture location (range = 0–6070 m). All resightings except seven (99.9 ± 0.05%; $\bar{x}$ ± 1 SE) were within 4.5 km of their initial capture location, while 70.8 ± 0.6% ($\bar{x}$ ± 1 SE) of resightings were within 1 km of their initial capture location (Fig 2).

### Landscape scale habitat use

At the landscape scale, we observed that SBTF strongly preferred three vegetation communities (Fig 3). These preferences were for *Eucalyptus melanophloia* woodland (odds ratio [OR] = 22.8, 95% confidence interval [95% CI]: 8.06–109), *E. brownii* woodland (OR = 17.5, 95% CI: 6.14–83.9) and *E. similis* woodland (OR = 17.7, 95% CI: 6.24–84.8). Conversely, SBTF strongly avoided non-remnant vegetation (OR = 0.021, 95% CI: 0.00015–0.259). There was little seasonal difference in the use of most vegetation communities (Fig 3). However, there was a weak but statistically significant decrease in the use of *E. brownii* woodland and *E. similis* woodland

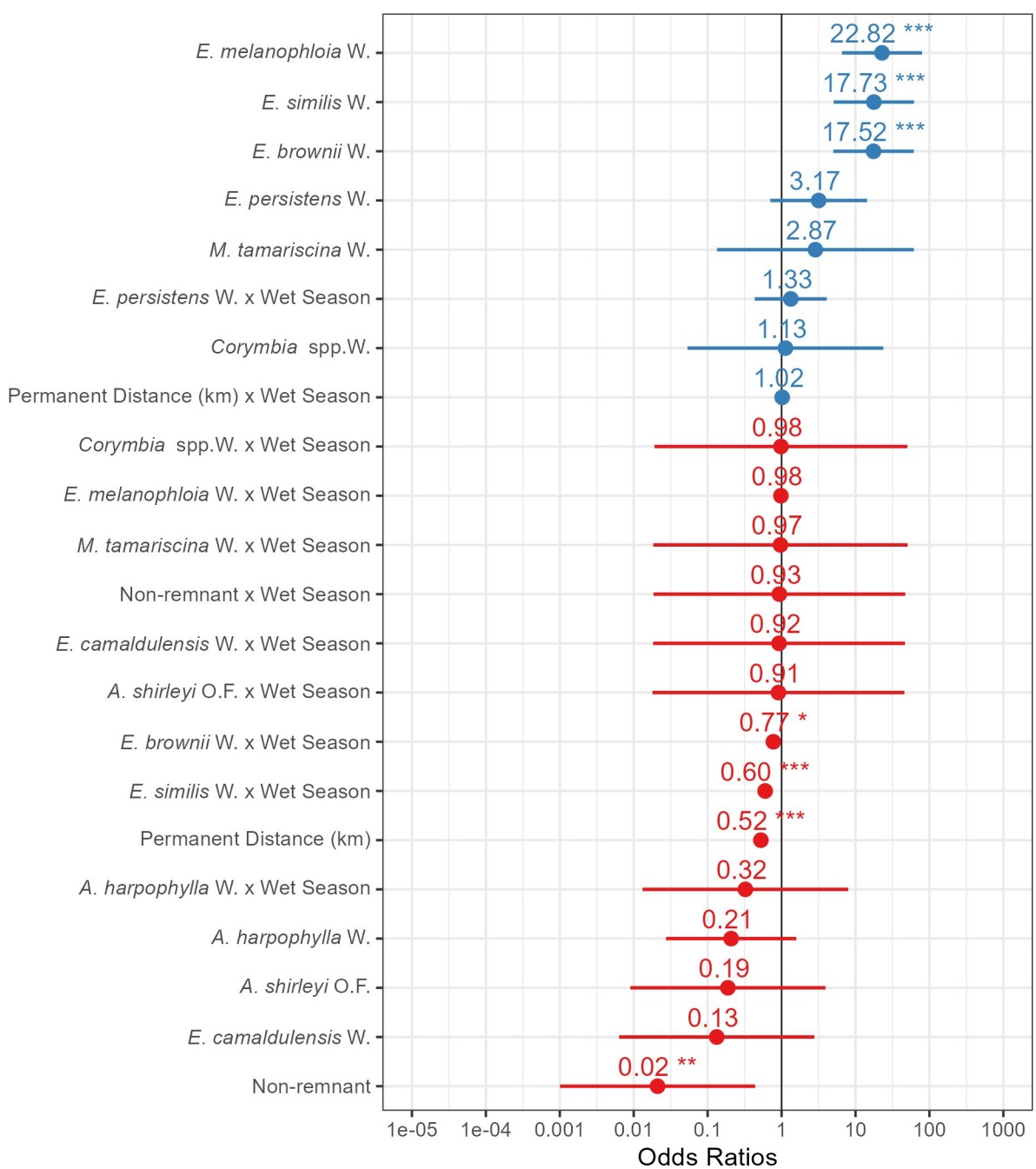

**Fig 3. Odds ratios of the predictor variables of a logistic regression resource use model in the Desert Uplands region of Queensland, Australia.** Odds ratios above 1 represent a use preference by Southern Black-throated Finch (blue), whereas odds ratios lower than 1 represent avoidance (red). The error bars show 95% confidence intervals. ***p < 0.001, **p < 0.01, *p < 0.05. Abbreviations include A = *Acacia*, E = *Eucalyptus*, M = *Melaleuca*, O.F. = open forest and W = woodland.

during the wet season compared to the dry season (OR = 0.774, 95% CI: 0.630–0.949; OR = 0.599, 95% CI: 0.499–0.719; respectively; Fig 3).

We resighted radio tracked SBTF at an average of 1240 ± 868 m from permanent water (range = 5.34–4210 m). Southern black-throated finch habitat use was strongly related to the distance to permanent water, with SBTF preferentially using areas closer to permanent water

(OR = 0.525, 95% CI: 0.498–0.553; Fig 3). Their use of these areas did not differ significantly among seasons (OR = 1.02, 95% CI: 0.945–1.10; Fig 3).

## Home range

We were able to fit utilisation distributions (UDs) in ctmm for 121 SBTF that had a sufficient effective sample size [48, 54]. The average tracking period for these UDs was 21.8 ± 22.5 days (range = 0.827–120 days), and the average number of positions per individual was 42.4 ± 33.1 (range = 5–199).

The mean 95% AKDE home range estimate was 319 ± 43.2 ha ($\bar{x}$ ± 1 SE, range = 8.58–3210 ha) and 50% core range estimate was 72.4 ± 10.2 ha ($\bar{x}$ ± 1 SE, range = 1.89–752 ha). The mean MCP home range estimate was 176 ± 17.2 ha ($\bar{x}$ ± 1 SE, range = 2.10–1220 ha). There was little difference in the 95% home range sizes of SBTF between the wet and dry seasons (GLMM: β = 0.27, 95% CI: -0.17–0.71, S3 Table). Furthermore, the distribution of SBTF home ranges within the study area was broadly similar between the dry and wet seasons (S2 Fig).

The 95% and 50% AKDE home range areas did not significantly correlate with the effective sample size (linear regression: $R^2$ = 0.01, n = 121, p = 0.118; and $R^2$ = 0.0007, n = 121, p = 0.30; respectively; S3 Fig), indicating that the autocorrelation and small sample size correction measures employed were successful in mitigating sample size bias in home range estimates.

## Diel activity patterns

We recorded 1054 SBTF activity observations through manual radio tracking, of which 990 were during diurnal hours, between 6:30 am and 5:30 pm (solar corrected times). Most of these records were SBTF foraging (48.0%, n = 480) or a combination of perching, preening or flying (42.8%, n = 426). Observations of drinking (4.66%, n = 46) and birds within nests (4.55%, n = 47) were less frequent. All nocturnal observations (6:30 pm to 5:30 am) were of SBTF within nests (n = 27; Fig 4).

Dawn and dusk radio tracking indicated that SBTF typically leave their nests around first light approximately 30 minutes before sunrise. In the evening, they engaged in extended perching and preening sessions before moving close to their nests at sunset and entering them around last light. Activities observed at dawn and dusk were mixed and included perching, preening, and flying (51.4%, n = 19), foraging (21.6%, n = 8) and birds within nests (27%, n = 10) (Fig 4).

There were significant diurnal patterns in SBTF activities (Fig 5). The early morning and late afternoon periods had similar proportions of foraging (β = -0.071, 95% CI: -0.17–0.026) and perching, preening and flying (β = 0.059, 95% CI: -0.038–0.16) (Fig 5 and S4 Table). Compared to the early morning, there were significantly lower proportions of foraging in the late morning (β = -0.19, 95% CI: -0.28–-0.09), midday (β = -0.18, 95% CI: -0.28–-0.085) and early afternoon (β = -0.26, 95% CI: -0.36–-0.17). In contrast, perching, preening and flying was significantly higher in the late morning (β = 0.13, 95% CI: 0.028–0.22), midday (β = 0.13, 95% CI: 0.35–0.23) and early afternoon (β = 0.24, 95% CI: 0.15–0.34) than compared to the early morning and late afternoon (Fig 5 and S4 Table).

Drinking was most common at midday (β = 0.073, 95% CI: -0.024–0.17); however, the confidence intervals overlapped with the late morning (β = 0.038, 95% CI: -0.059–0.13), early afternoon (β = 0.036, 95% CI: -0.061–0.13) and late afternoon (β = 0.046, 95% CI: -0.051–0.14). No drinking observations were recorded during the early morning period.

The only diurnal activity exhibiting seasonal variation was records of birds within nests, with these being significantly higher in the wet season than the dry season (95% CI: 0.0041–0.13; S4 Table).

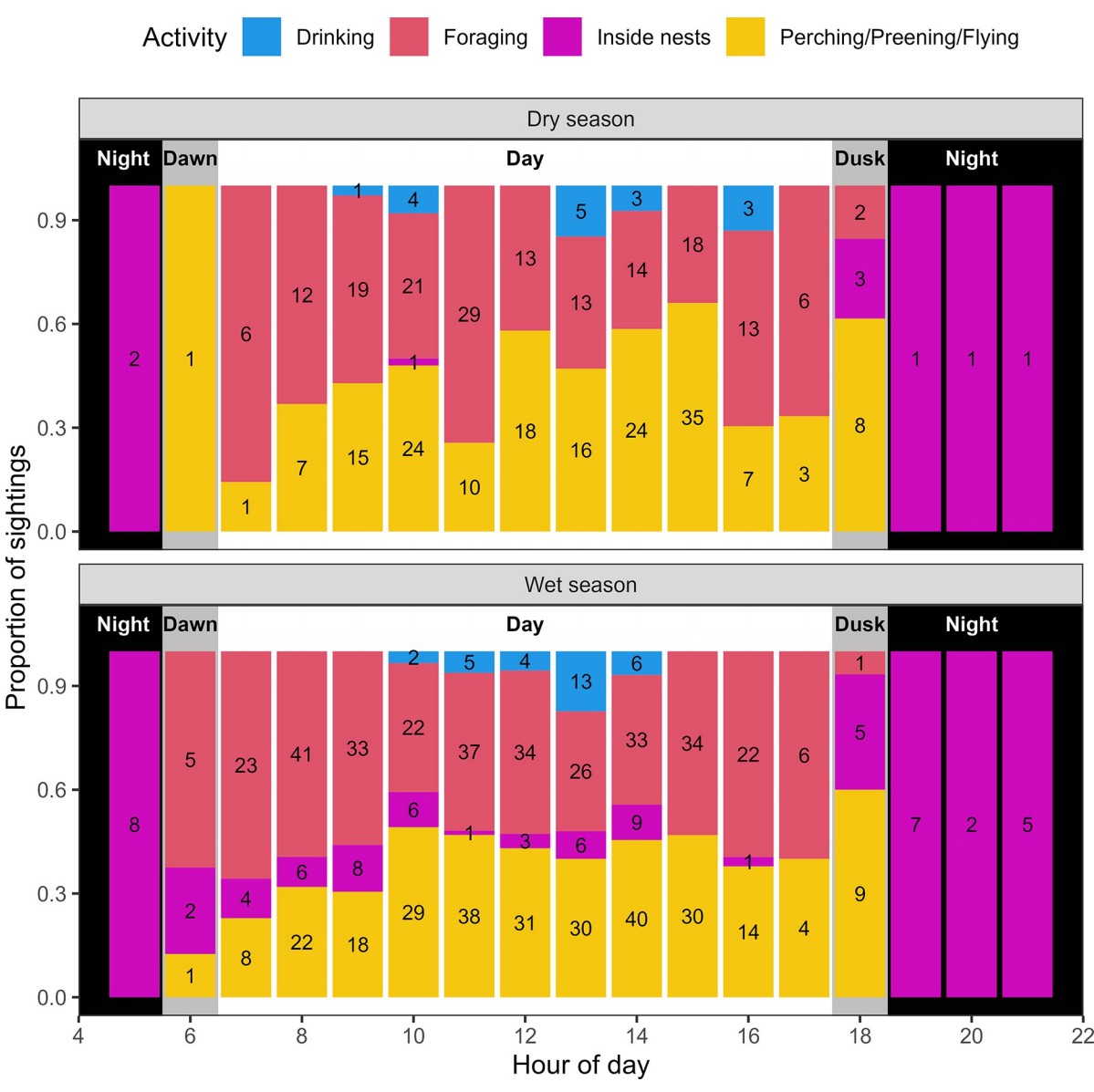

**Fig 4. Daily activity records of Southern Black-throated Finch by time of day and season.** Numbers show a count of the number of observations for each bar.

## Discussion

### Landscape scale habitat use

Resource availability is a key driver of habitat use [33, 55]. The distribution of preferred vegetation communities and drinking sites is therefore expected to affect habitat use by granivorous birds. We found that at a landscape scale, SBTF habitat use was strongly associated with three vegetation communities and the availability of permanent water. The SBTF strongly selected woodlands dominated by *E. melanophloia*, *E. brownii* and *E. similis*, which within the study area comprise habitats with a high richness of native tussock grass species that comprise the majority of the SBTF diet [18, 56, 57]. Conversely, non-remnant vegetation was strongly avoided. Within the study area, non-remnant vegetation is dominated by buffel grass

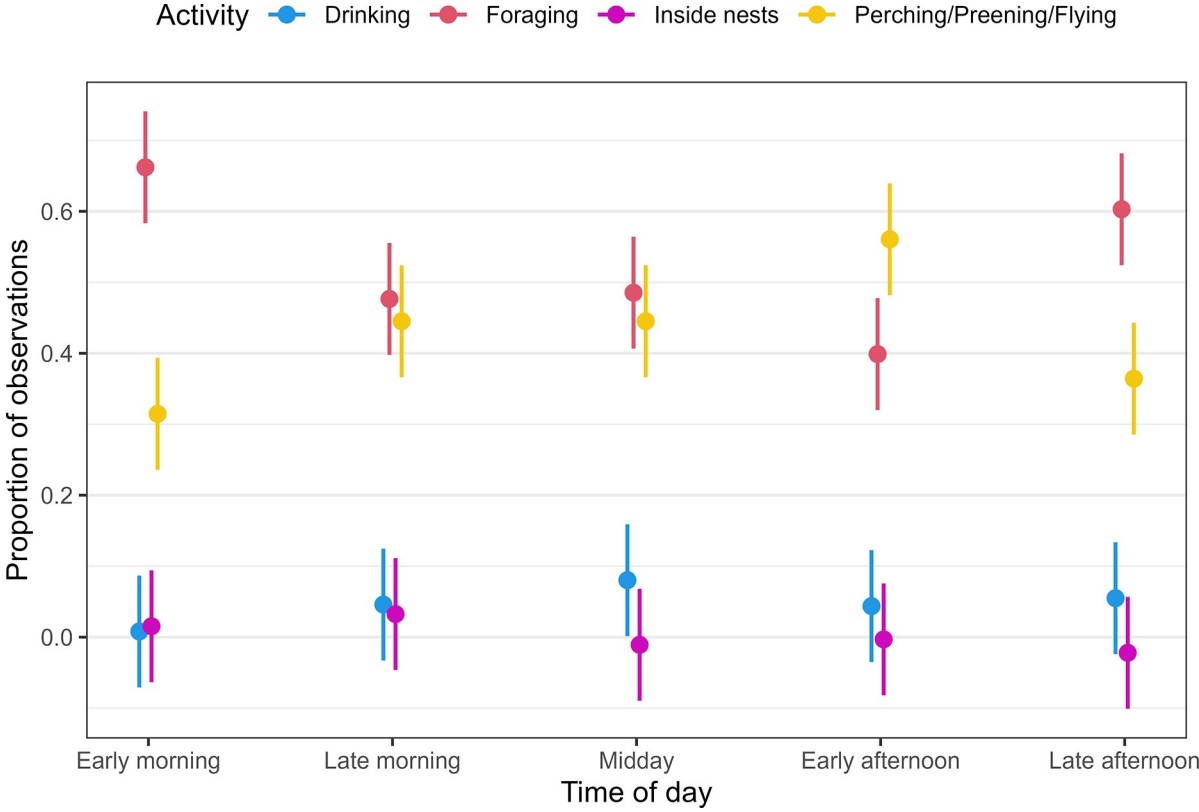

**Fig 5. Diel activity patterns of Southern Black-throated Finch.** The bars show confidence intervals established using a linear model. Dusk and night periods were excluded from the linear model due to sample size constraints.

(*Cenchrus ciliaris*), an introduced species that outcompetes native grasses and forbs [58] and provides little forage value for SBTF [59]. In addition, non-remnant areas have been historically cleared and lack canopy trees in which SBTF typically roost, nest and escape predators [52]. At the landscape scale, SBTF also preferred sites close to permanent water, suggesting that the species requirement for regular access to drinking water promotes the use of habitats requiring less travel costs. Close associations with permanent water have been identified for the SBTF population of the Townsville Coastal Plains, with Rechetelo et al. [6] resighting the majority of radio tracked SBTF within 200 m of water. While our study found a significant increase in SBTF use of habitat near water, the effect size was smaller than that suggested by Rechetelo et al. [6], with SBTF being resighted on average 1.2 km and up to 4.2 km from water. Our findings clearly demonstrate the importance of identifying the habitat preferences of threatened species, in order to target these habitats for conservation actions [3].

The availability of resources varies through time and space across the Australian savanna woodlands, primarily driven by variable rainfall, habitat and land management [15, 17]. Many granivorous birds in these areas track resources seasonally in response to this variation [60–62]. At the landscape scale, we found little seasonal change in SBTF habitat use. A small but statistically significant shift away from *E. brownii* and *E. similis* woodlands during the wet season was recorded, which suggests that SBTF may concentrate in *E. melanophloia* woodlands at times of high seed abundance and spread their activity over a wide range of vegetation in times of relative scarcity. However, broadly, this study suggests that at the landscape scale habitat use by SBTF remains largely similar across seasons. At least on short timeframes, of less than one

month, SBTF movement at the regional scale is also limited, with almost all radio tracked bird resightings occurring within 4.5 km of the initial capture location. Our results therefore suggest that the majority of SBTF resource tracking occurs on the finer local scale, which is the use of foraging, nesting and drinking sites within an individual's home range over hourly to daily timeframes [6, 33].

The scale of habitat use is an important factor in the conservation management of threatened species [63]. Conservation strategies must occur on spatial scales that are aligned with the species habitat use and resource requirements [3]. For instance, in savanna woodlands, best practice fire management regimes are characterised by patchy and low-intensity fires, which create a fine-scale mosaic of fire histories [64]. This approach is particularly advantageous for relatively sedentary granivores, like the SBTF, that depend on a diverse array of grass species within their local habitats to meet their nutritional needs throughout the year [7].

## Home range

Our study identified that SBTF predominately exhibit sedentary behaviour with largely stationary home ranges over short timeframes (less than one month). This finding contributes to our understanding of niche separation among sympatric estrildid finches, which are largely differentiated based on differences in their diet and mobility [12, 13]. Notably, the SBTF's home range and movement characteristics are similar to the closely related and morphologically similar Long-tailed Finch (*Poephila acuticauda*). Like SBTF, Long-tailed Finch occupy a similar habitat niche in the savannas of northwestern Australia [65], have large mostly sedentary home ranges and have similar diel drinking patterns [66]. In addition, SBTF share similar wing morphology to Long-tailed Finch, particularly the wing width-length ratio [67], which is a strong indicator of mobility among estrildid finches [12]. Collett et al. [13] reported that Long-tailed Finch diversified their diet in times of resource restriction, which was in contrast to the findings of the sympatric Gouldian Finch, a dietary specialist that uses regional nomadism to meet its resource requirements [60, 68]. The home range patterns and wing morphology of SBTF suggest a similar mobility niche to that of Long-tailed Finch, which is characterised by local-scale resource tracking within large home ranges (on average 319 ha) and limited broader landscape or regional movements [12, 13].

A sedentary home range pattern of SBTF with possible diet diversification in times of food scarcity is also supported by our finding that home ranges were stable across seasons. Rechetelo et al. [6] similarly reported that there was no significant seasonal variation in home range size of SBTF in the Townsville Coastal Plains. The diet of SBTF comprises a wide variety of grass seeds and a minor dietary component of insects when available [20, 57]. This dietary flexibility suggests that SBTF are likely to use dietary diversification within their local home range rather than broader landscape scale movements to meet their nutritional needs throughout the year [13].

Theory predicts that home range size is proportional to the availability and quality of suitable habitat, with larger home ranges required for areas with a sparser distribution of resources [69–71]. We recorded mean AKDE home range and core range estimates that were 6–7 times greater than those recorded by Rechetelo et al. [6] for the Townsville Coastal Plains population. In addition, our MCP home range estimates obtained using a simple and comparable method were 3.8 times greater than those recorded by Rechetelo et al. [6]. The observed difference in home range areas may be a function of resource density [69]. Compared to the Townsville Coastal Plains, the Desert Uplands has a sparser distribution of permanent water sources [72], which may require SBTF to extend their home range to gain water [73]. The Desert Uplands also has a lower annual rainfall, which may reduce the seed production of grassy woodlands

[74]. These two environmental differences may require SBTF to have a larger home range to fulfil their resource requirements [69, 71].

While the methods used to estimate MCP home ranges were consistent between this study and Rechetelo et al. [6], we adopted a more contemporary kernel density estimation (KDE) home range method, which was specifically designed to correct biases introduced by autocorrelated data and small sample sizes [46, 48]. Fleming et al. [48] used simulation tests to demonstrate that small effective sample sizes, which are common in radio tracking datasets, bias KDE home range estimates lower than true values. The autocorrelation and small sample size weighting methods that we used may explain why our KDE estimates were 6–7 times greater than those of Rechetelo et al. [6], while the MCP estimates were less than four times greater.

## Diel activity patterns

Our results revealed distinct diurnal activity patterns in SBTF. There was a bimodal peak in SBTF foraging activity, with increased foraging in the early morning and late afternoon. A bimodal peak in forging activity has been identified for other Australian estrildid finches [52]. However, SBTF specific studies have reported varying results. Using 659 activity observations of SBTF, Mitchell [56] did not find a daily pattern to foraging activity, while in contrast, a coarse analysis of 620 activity observations by Rechetelo et al. [6] suggested a pattern of bimodal foraging similar to our study. For tropical granivorous finches, this bimodal pattern has been suggested to be driven by energy and water conservation requirements during the hottest periods of the day [75, 76].

Drinking activities were highest around the midday period, followed by lower rates of drinking in the late morning, early afternoon, and late afternoon; however, the rates of drinking were not significantly different from the early morning observations in the linear model. This lack of significance may be due to a limited sample size of drinking observations through manual tracking. The timing of drinking in estrildid finches is species-specific [66, 77]. Gouldian Finch and Pictorella Mannikin favour drinking in the morning, while Double-barred Finch, Masked Finch (*Poephila personata*), Long-tailed Finch, Zebra Finch (*Taeniopygia castanotis*) and Star Finch (*Bathilda ruficauda*) have been found to drink throughout the day, sometimes with peak visitation rates occurring during the middle of the day [66, 77]. Although we did not find a statistically significant pattern in SBTF visitation of water sources, the diurnal timing of our drinking records supports the temporal spread of drinking behaviour in SBTF that is likely to be concentrated around the hottest parts of the day. We also found a concentration of SBTF resting (perching and preening) in the middle of the day, which was also reported by Zann [52] for closely related estrildid finches.

The timing of departure from the nest near first light (approximately 30 minutes prior to sunrise) and re-entry to the nest at last light (approximately 30 minutes after sunset) aligns with previous ethological research on SBTF [52]. Nest visitation was nearly only recorded at night during the non-breeding dry season, while during the wet season, there was an increased rate of nest visitation during the day, aligning with daytime nesting activities, such as incubation, occurring during the breeding season [20, 52].

## Conclusion and conservation recommendations

Our study identified strong landscape scale habitat selection by SBTF for grassy eucalypt woodlands where permanent water was accessible and avoidance of non-remnant habitat. This study revealed that SBTF occupy large but sedentary home ranges within which they track resources at a local scale, with limited landscape scale movements, at least over timeframes of less than a month. While our results report on the responses for a single threatened

species, these have implications for other resource tracking species more broadly. Importantly, our results highlight the need for local scale habitat management that protects preferred habitats and supports a diversity of suitable forage plants required by the target species.

We found that SBTF foraging activity was most concentrated in the early morning and late afternoon. In addition, while there was no significant effect of time-of-day on drinking behaviour, the highest rates of drinking were observed during the heat of day. Current survey guidelines for SBTF recommend water source monitoring for at least three hours after first light and targeted searches for foraging individuals within close proximity (600 m) of water sources [78]. Our findings recommend that the SBTF survey guidelines be updated to include water source watches during the middle of the day and targeted searches for foraging SBTF in the early morning and late afternoon.

## Supporting information

**S1 Fig. Distribution of presence sites and pseudoabsence sites relative to their distance to the nearest vehicle track.**
(TIF)

**S2 Fig. Map of the study area showing the locations of presence and pseudoabsence sites as well as mean wet and dry season SBTF home ranges.**
(TIF)

**S3 Fig. 95% autocorrelated kernel density estimate (AKDE) home ranges and 50% AKDE core ranges compared to the effective sample sizes that are calculated within the ctmm package.**
(TIF)

**S1 Table. Treatment information and home range sizes of SBTF.**
(XLSX)

**S2 Table. Vegetation communities within the study area.**
(DOCX)

**S3 Table. Estimated regression parameters, standard errors, z-values and p values for a generalised linear mixed effects model relating home range size to season.**
(DOCX)

**S4 Table. Estimated regression parameters, standard errors, t-values and P-values for the linear model assessing the relationship among activity, season and time of day.**
(DOCX)

## Acknowledgments

We gratefully thank Joshua Moore, Lyndall Marshall, Andy Page, Melinda Bergmann, Samuel Wilson, Ed Meyer, Dean Jones, Chays Ogston, Jessica Hogg and Breanna Williams for their administrative, technical and field work support for the project, and we are indebted to their efforts. We also thank Juliana McCosker for long-term support of the project and of SBTF conservation broadly.

## Author Contributions

**Conceptualization:** John M. van Osta, Brad Dreis, J. Guy Castley.

**Data curation:** John M. van Osta.

**Formal analysis:** John M. van Osta.

**Funding acquisition:** Brad Dreis.

**Investigation:** John M. van Osta.

**Methodology:** John M. van Osta.

**Project administration:** Brad Dreis.

**Software:** John M. van Osta.

**Supervision:** Laura F. Grogan, J. Guy Castley.

**Validation:** John M. van Osta.

**Visualization:** John M. van Osta.

**Writing – original draft:** John M. van Osta.

**Writing – review & editing:** John M. van Osta, Laura F. Grogan, J. Guy Castley.

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
