## [Decision Letter · Decision Letter 0]

2 Jun 2024

PONE-D-24-14262Local resource availability drives habitat selection by a threatened avian granivore in savanna woodlandsPLOS ONE

Dear Dr. van osta,

Thank you for submitting your manuscript to PLOS ONE. After careful consideration, we feel that it has merit but does not fully meet PLOS ONE’s publication criteria as it currently stands. Therefore, we invite you to submit a revised version of the manuscript that addresses the points raised during the review process.

We look forward to receiving your revised manuscript.

Kind regards,

Dárius Pukenis Tubelis, Ph.D.

Academic Editor

PLOS ONE

Journal Requirements:

"Funding provided by Bravus Mining and Resources"

"We gratefully thank Bravus Mining and Resources for their funding and support. Many staff members of E2M Pty Ltd provided administrative, logistical and field work support for the project, and we are indebted to their efforts. We also thank Juliana McCosker for long-term support of the project and of SBTF conservation broadly"

"Funding provided by Bravus Mining and Resources"

5. In the online submission form, you indicated that your data is available only on request from a third party. Please note that your Data Availability Statement is currently missing contact details for the third party, such as an email address or a link to where data requests can be made. Please update your statement with the missing information. 

**Additional Editor Comments:**

Dear Dr John Michael van Osta,

Thank you for submitting your manuscript on the habitat of the SBTF to PLOS ONE.

This submission received the code PONE-D-24-14262 in our system.

We received two external reviews and I also reviewed your manuscript.

I took a bit long to send this decision because I was waiting the opinion of a third external reviewer, who has not sent the review. I do not want to wait more.

Reviewer 1 suggested Minor Revision and provided some compliments. She/he provided a set of suggestions concerned with the Methods section.

Reviewer 2 suggested Minor Revision and provided a range of compliments regarding the quality and importance of your study. Her/his comments and suggestions refers mainly to the need of more clarity in the Methods section.

Both reviewers mentioned the lack of availability of your data in the initial section of questions. Maybe, you can add some basic data as Supplementary Material. Please check Instructions about this again to verify what is better for you, but informative and important for authors.

Please find my own review below.

Please feel free to contact me if you have any doubt regarding this submission.

Dárius P. Tubelis

PLOS ONE Editor

Additional review by the editor (Dárius P. Tubelis):

Title.

I´m a bit worried with the use of the term "selection". Have you examined it ? Ususally, selection involves the measurement of habitat use and availability. Please talk

with the co-authors again about this and follow what is better for you.

Introduction

Good lenght, good content, congratulations.

Formatting

You are using round brackets to cite references. It is wrong.

You have to use square brackets: So, use, for example, [2] instead of (2). Here and along all parts of the text.

Also, when you cite two references, you have to add a space after the comma. Then you will have [2, 3].

Also, you are using hyphen to separate two consecutive numbers of references, but you have to use a long dash. Please see recent papers for comparison.

Line 85-94. Paragraph of the objective. Please note that here you use "habitat use", not selection. Have to check and decide.

Material and Methods

Line 98. You have to use "(Fig 1)". Abbreviated and without dot. Please check this for all figures along the paragraphs.

Line 106. Add 1-2 refs at the end.

Line 109. I think the correct is "(Fig 1, S2 Table).

Line 177-119. Use "Fig 1." in bold in captions. It is better if you complete the first sentence by citing the full species name, savannas, the Australian region.

Line 137. Use just "(S1 Table)". Please follow this for all tables and figures as supplementary material.

Lines 173-174. "towers" appears four times in two lines. Could you make some changes to avoid these repetitions ?

Results.

Line 302. Use "Fig 2." in bold in captions. Add the Australian region at the end of the first sentence.

Line 306. Use "(Fig 3).". Please chekc this for all figures.

Line 320. Inform the Australian region. Also, you have to explain in the caption what are these scientific names, O.F. and W..

Explain the red and blue colors. Delete the SBTF abbreviation after the full name.

Line 375. Figure 5. Some people will not distinguish the red and green colors. Consider changing some colors here and in other figures.

Discussion

It appears to be well divided in sections and very well written.

References

They are partly formatted. I think you have to do the following changes:

After the year, you must have a semi-colon and a space "2021; 15:"

You must have a space prior to the pages, and they have to be separated by a long dash, not hyphen.

It is better if you add the DOI for all the refs that have it.

Please check the Instructions again.

Dárius

Reviewers' comments:

Reviewer's Responses to Questions

**Comments to the Author**

1. Is the manuscript technically sound, and do the data support the conclusions?

Reviewer #1: Yes

Reviewer #2: Yes

2. Has the statistical analysis been performed appropriately and rigorously? 

Reviewer #1: Yes

Reviewer #2: Yes

3. Have the authors made all data underlying the findings in their manuscript fully available?

Reviewer #1: No

Reviewer #2: No

4. Is the manuscript presented in an intelligible fashion and written in standard English?

Reviewer #1: Yes

Reviewer #2: Yes

5. Review Comments to the Author

Reviewer #1: The manuscript is well written and contains important information about an Australian endemic species. The manuscript presents appropriate background context and related research. Suggestions to improve overall understanding are provided below.

Line 22 - state the number of towers used

Line 87-88 - perhaps move this information to the methodology section

Line 101-102 - if possible, add mining infrastructure, major roads, Carmichael River, and other features that may influence habitat selection to the map

Line 115 - if possible add average distance of mining infrastructure to ART towers or acknowledge distance of banding stations/towers to mining area.

Line 116 - clarify if non-remnant is all mining area or if it's something else

Line 143 - explain ART acronym

Line 132 - was the number of BTF banded the same as the number of BTF tagged?

Line 151 - add details on how many BTF were manually tracked, how many BTF were tracked for 5hs, etc.

Line 171 - "throughtout" gives the idea that towers were homogeneously distributed and not grouped. Rewrite to clarify and add information about what vegetation community the towers were placed in.

Line 468 - Rechetelo et al. did a rough analysis of the activity patterns on the ground, maybe you can confront the information.

Reviewer #2: Authors present a solid study on the habitat use of radio-collared black-throated finches evaluated across three seasons and replicated across three years. The study found that this species has large home ranges to include food resources that vary in space and time. Overall, I very much enjoyed reading the paper as it is well written, easy to follow, and is based on a robust study design and analytical approach. The results of the paper will provide important information to develop sound conservation strategies.

Given the excellent quality of the paper, I only have a few minor comments and edits.

General comments:

1. Please revise the manuscript to increase use of active voice.

2. I suggest a careful revision of word choice when describing habitat use vs selection. To my understanding of your sampling and analytical approach, your paper evaluates habitat use of black-throated finches and not selection as you are not analytically determine use versus availability of habitat types (i.e., land-cover types). As a result, you need to replace “selection” with “use”. If my understanding is incorrect, please pardon the oversight, then you need to add text describing in detail that your study evaluates habitat selection and not use.

3. Please provide a more detailed description of ecosystem types for readers who have not had the opportunity to experience them.

4. When did you commence data collection after capture? Did you use the commonly applied 24-hr waiting period?

5. Please provide links to R code and raw data for readers to replicate analyses. If locations of black-throated finches cannot be made public due conservation concerns, please state so.

Minor comments:

Line 26: Provide range of tracking periods. The SD is not informative other than to demonstrate that there is a lot of variation in tracking periods.

Line 40: Replace “selection choice” with “use”.

Line 77: Refer Fig. 1 here as readers unfamiliar with the geography of Australia may not know the location of the Townsville Coastal Plains.

Line 147: SD format differs from above.

Line 206: Describe the process of how you selected two presence sites per day? Did you use a stratified approach where observations were separated by a certain length of time or restricted to morning vs evening?

Line 209: Explain your rationale for increasing pseudo-absence points by a factor of 10 compared number of presence sites.

Line 246: Provide criteria when subsampling ART locations.

Line 286: Provide general statement here stating that data are reported as mean +/- SD (range). This will decrease repeated verbiage in the Result section.

Line 386: Define “non-remnant” in Method section.

Figure 3: See general comment 3. Also color legend to text: … used (blue)

6. PLOS authors have the option to publish the peer review history of their article (what does this mean?). If published, this will include your full peer review and any attached files.

Reviewer #1: No

Reviewer #2: **Yes: **Matthias Leu

---

## [Author Response · Author response to Decision Letter 0]

19 Jun 2024

Thank you for your constructure review. We have accepted and addressed your comments in full, as detailed within the attached 'Response to Reviewers.docx' file.

---

## [Editor Report · Decision Letter 1]

24 Jun 2024

Local resource availability drives habitat use by a threatened avian granivore in savanna woodlands

PONE-D-24-14262R1

Dear Dr. John M. van Osta,

We’re pleased to inform you that your manuscript has been judged scientifically suitable for publication and will be formally accepted for publication once it meets all outstanding technical requirements.

Kind regards,

Dárius Pukenis Tubelis, Ph.D.

Academic Editor

PLOS ONE

Additional Editor Comments:

Dear Dr John M. van Osta,

Thank you for submitting the corrected version of your manuscript on habitat use by Poephila cincta cincta in Australia (PONE-D-24-14262R1).

I agree with your responses and actions regarding the comments and suggestions provided by both reviewers and me.

I noted that you followed all of them.

As the changes on the manuscript were quite appropriate, it has been improved.

I consider that your submission now meets the PLOS ONE publication criteria.

Thus, I suggest Acceptance.

PLOS ONE people might contact you along the next days for final actions.

During my last reading of the manuscript I found a small set of minor errors that have to be fixed prior to publication.

Please find these errors and suggested corrections below, and follow then prior to or during the proofs correction.

Thank you for considering PLOS ONE as home of your research.

Dr. Dárius P. Tubelis

PLOS ONE Editor

**Final errors to be fixed by authors:**

Line 71. Maybe you can replace "The species habitat" by "Its habitat". Note that you are repeating the same term in consecutive sentences, one below the other.

Line 78. I suggest you delete "(Fig 1)". I consider strange citing your own figure in the Introduction. You already cited two references...

Line 88. I´m used to see "et al." with no italics in PLOS ONE papers. Please check again if the italics is necessary.

Lines 198 and 201. The same.

Line 209. Is "BTF" correct ?

Line 219. Et al. Please search for all along the text and eliminate the italics is necessary. I think so.

Line 220. I consider strange having two STBF on the same sentence. Maybe you can replace the second one by "these birds" or similar.

Line 226. A space is lacking between the round and square brackets.

Line 509. Two SBTF. Can you replace the second one by a pronoum or other word ?

The references appear to be well formatted.

All figures and tables were cited along the text.

However, I did find links to the Supplementary Materials....

Good work!!

Dárius

---

## [Editor Report · Acceptance letter]

12 Jul 2024

PONE-D-24-14262R1 

PLOS ONE

Dear Dr. van osta, 

I'm pleased to inform you that your manuscript has been deemed suitable for publication in PLOS ONE. Congratulations! Your manuscript is now being handed over to our production team.

Kind regards, 

on behalf of

Dr. Dárius Pukenis Tubelis 

Academic Editor

PLOS ONE